# Regulation of VEGFR2 and AKT Signaling by Musashi-2 in Lung Cancer

**DOI:** 10.3390/cancers15092529

**Published:** 2023-04-28

**Authors:** Igor Bychkov, Iuliia Topchu, Petr Makhov, Alexander Kudinov, Jyoti D. Patel, Yanis Boumber

**Affiliations:** 1Robert H Lurie Comprehensive Cancer Center, Northwestern University, Chicago, IL 60611, USA; 2Program in Molecular Therapeutics, Fox Chase Cancer Center, Philadelphia, PA 19111, USA; 3Cardiology Department, University of Illinois in Chicago, 840 S. Wood Street, Chicago, IL 60612, USA; 4Division of Hematology/Oncology, Section of Thoracic Head and Neck Medical Oncology, Feinberg School of Medicine, Northwestern University, Chicago, IL 60611, USA

**Keywords:** non-small cell lung cancer, vascular endothelial growth factor receptor-2, Musashi-2, PTEN

## Abstract

**Simple Summary:**

Lung cancer is the most common and lethal malignancy worldwide. Musashi-2 (MSI2) is an RNA-binding protein that is overexpressed in advanced NSCLC. VEGFR2 protein expression contributes to NSCLC progression and several FDA-approved drugs are used to target it in the clinic. Here, we show that MSI2 is a strong positive regulator of VEGFR2 protein levels in murine and human NSCLC cell lines. Furthermore, we found that MSI2 protein directly binds to *VEGFR2* and *PTEN* mRNAs and impacts VEGFR2 downstream signaling, in part via PTEN regulation.

**Abstract:**

Lung cancer is the most frequently diagnosed cancer type and the leading cause of cancer-related deaths worldwide. Non-small cell lung cancer (NSCLC) represents most of the diagnoses of lung cancer. Vascular endothelial growth factor receptor-2 (VEGFR2) is a member of the VEGF family of receptor tyrosine kinase proteins, which are expressed on both endothelial and tumor cells, are one of the key proteins contributing to cancer development, and are involved in drug resistance. We previously showed that Musashi-2 (MSI2) RNA-binding protein is associated with NSCLC progression by regulating several signaling pathways relevant to NSCLC. In this study, we performed Reverse Protein Phase Array (RPPA) analysis of murine lung cancer, which suggests that VEGFR2 protein is strongly positively regulated by MSI2. Next, we validated VEGFR2 protein regulation by MSI2 in several human lung adenocarcinoma cell line models. Additionally, we found that MSI2 affected AKT signaling via negative *PTEN* mRNA translation regulation. In silico prediction analysis suggested that both *VEGFR2* and *PTEN* mRNAs have predicted binding sites for MSI2. We next performed RNA immunoprecipitation coupled with quantitative PCR, which confirmed that MSI2 directly binds to VEGFR2 and PTEN mRNAs, suggesting a direct regulation mechanism. Finally, MSI2 expression positively correlated with VEGFR2 and VEGF-A protein levels in human lung adenocarcinoma samples. We conclude that the MSI2/VEGFR2 axis contributes to lung adenocarcinoma progression and is worth further investigations and therapeutic targeting.

## 1. Introduction

Lung cancer is second in incidence and first in mortality among all cancer types in recent years based on World Health Organization and American Cancer Society reports [1,2]. Most lung cancer statistics include both small cell lung cancer (SCLC) and non-small cell lung cancer (NSCLC). In general, NSCLC represents about 85% of all lung cancer and includes squamous (30%) and non-squamous (70%) subtypes [3,4,5]. Lung cancer typically has a poor prognosis if tumors have disseminated from the primary site [6]. Besides the most common oncogenic driver mutations, *KRAS* and *TP53*, NSCLC is associated with numerous genetic alterations such as *EGFR* mutations, *ALK* translocation, *VEGFR2* and *c-MET* amplification, and others [7,8]. At this time, in addition to chemotherapy, immunotherapy, and targeted therapies, VEGFR2 and VEGF inhibitors have shown proven efficacy in NSCLC treatment in stage IV disease in combination with chemotherapy and also have also shown promise when combined with immunotherapy [3,9].

Vascular endothelial growth factor (VEGF-A) and its receptors (VEGFR1, VEGFR2, VEGFR3) are the main pro-angiogenic drivers of solid tumors [10]. In the canonical model, VEGF-A is expressed by tumors and recognizes and binds VEGFR2 on endothelial cells, thus leading to tumor vessel formation [11] and, as a result, a better supply of nutrients for tumor cells. Additionally, VEGF-A promotes an immunosuppressive tumor microenvironment via several mechanisms [12,13,14,15]. Published papers show that VEGFR2 and VEGFR1 are well expressed on tumor cells [16,17,18,19]. Moreover, *VEGFR2* gain-of-function is associated with chemoresistance and poor survival in patients with lung cancer [20]. Currently, several drugs are used in the clinic, including VEGFR2 inhibitors (ramucirumab, cabozantinib, pazopanib), a VEGF-A inhibitor (bevacizumab) [3], and a VEGF-A/PIGF trap (aflibercept) [21]. Adding the VEGFR2 inhibitor nintedanib to second-line docetaxel showed improved progression-free survival (median 3.4 months with combination therapy versus 2.7 months with docetaxel alone; HR 0.79, *p* = 0.0019) [22]. Additionally, a randomized phase III study compared a combination of ramucirumab with docetaxel to docetaxel alone and reported a modest improvement in median overall survival from 9.1 months to 10.5 months (HR 0.86, *p* = 0.023) [23].

Furthermore, Musashi-2 (MSI2) protein expression is associated with advanced NSCLC [24,25]. MSI2 and its homolog, Musashi-1 (MSI1), are a family of RNA-binding proteins that regulate the stability and translation of target mRNAs through highly specific RNA binding of the core motif in the 3′-untranslated region of mRNAs (UAG) [26,27,28]. MSI2 regulates multiple critical biological processes in stem cells and cancer cells and contributes to cancer drug resistance [29,30]. It was previously indicated that MSI2 protein is increased in liquid and solid tumors, of which lung cancer is one of them [24,27,31,32,33,34]. Recently, we showed that MSI2 protein level correlates with NSCLC aggressiveness and has a specific role in promoting metastasis in these tumors via regulation of TGFβR1 and its target SMAD3 [25]. Furthermore, we have recently shown that MSI2 deficiency leads to higher sensitivity to Epidermal Growth Factor Receptor (EGFR) inhibitors in EGFR mutant NSCLC due to EGFR protein downregulation resulting from MSI2-mediated direct regulation of EGFR mRNA [24].

In the light of published papers, we have validated proteomic analysis data that suggested VEGFR2 regulation upon MSI2 depletion in murine lung cancer. We found that MSI2 depletion leads to a strong decrease in VEGFR2 protein levels in murine and human lung adenocarcinoma cell lines. Furthermore, we found that MSI2 protein directly binds to *VEGFR2* and *PTEN* mRNAs and impacts VEGFR2 downstream signaling via PTEN regulation. Finally, MSI2 protein expression correlated with VEGFR2 and VEGF-A protein levels in NSCLC patient samples.

## 2. Materials and Methods

### 2.1. Cell Culture

Human lung adenocarcinoma cell lines A549 (*KRAS^34G>A^*, *CDKNA^1_471del471^*), Calu-1 (*KRAS^34G>T^*, *TP53^del^*), H23 (*KRAS^34G>T^*, *TP53^738G>C^*), and H441 (*KRAS^35G>T^*, *TP53^473G>T^*) were obtained from the American Type Culture Collection (ATCC). Human lung cancer cell lines Hcc1171 (*KRAS^34G>T^*, *TP53^740A>T^*) and Hcc461 (*KRAS^35G>A^*, *TP53^445delT^*) were obtained from UT Southwestern Medical Center. The murine NSCLC cell line (344SQ) from Trp53R172HΔG/+ KrasLA1/+ mice was previously described [35]. Initial stocks were cryopreserved and, at every 6-month interval, a fresh aliquot of frozen cells was used for the experiments. No additional authentication was performed. All cells were cultured in RPMI 1640 (Gibco, Gaithersburg, MD, USA) supplemented with 10% FBS (Hyclone, Logan, UT, USA), penicillin (100 U/mL), streptomycin (100 µg/mL), sodium pyruvate (1 mM), and non-essential amino acids (0.1 mM) under the conditions indicated in the figure legends. Hypoxic incubations were performed with 1% O_2_, 94% N_2_, and 5% CO_2._

### 2.2. Antibodies and Drugs

Anti-MSI2 (#ab76148) and anti-VEGF-A (#ab46154) were obtained from Abcam (Cambridge, UK). Anti-VEGFR2 (#2479), anti-phVEGFR2 (#3817), anti-β-actin (#3700), anti-phAKT (T308) (#13038), anti-phAKT (S473) (#4060), anti-AKT total (#2920), anti-phGSKαβ (#9331), anti-GSKαβ total (#5676), anti-PTEN (#14642), anti-phERK (#4370), anti-ERK total (4695), anti-ph4EBP(#2855), anti-phP70SK (#9206), anti-P70S6K total (#2708), anti-rabbit HRP-linked (#7074), and anti-mouse HRP-linked (#7076) were obtained from Cell Signaling (Danvers, MA, USA). Doxycycline (#HY-N0565), hygromycin (HY-B0490), and puromycin (HY-B1743A) were obtained from MedChemExpress (Monmouth Junction, NJ, USA).

### 2.3. Vector Construction and Lentivirus Production

To generate cell lines with inducible knockdown of MSI2, we used self-complementary single-stranded DNA oligos from Appendix A as shown in our published paper [24]. To generate stable cell lines with KDR (VEGFR2) overexpression, we used pHage-KDR-hygro vector and pHage-hygro vector as negative control. We used plasmids pHage-KDR (#116754) and pHage-puro (#118692) from Addgene with a changed resistance gene to hygromycin using a cDNA containing a Hygro gene and cloned into ClaI/NdeI and BfuAI/ClaI sites to generate overexpression constructs. The pLV-CMV-hygro vector used as a template for the Hygro gene. All generated cell lines used in the study are noted in Appendix A.

### 2.4. SiRNA Transfections

SiRNA transfection was performed using nonspecific control pool siRNAs and anti-human MSI2 SiRNAs (Appendix A). Details of the procedure were previously described in our published paper [24].

### 2.5. Western Blot Analysis

Cell lysate preparation and Western blot analysis were performed using standard methods as previously described [1]. Band signals were detected using X-ray film, and the film was digitized using a photo scanner. Image analysis was conducted using ImageJ (version 1.53e, National Institutes of Health, Bethesda, MD, USA) with signal intensity normalized to β-actin, and 3–4 repeats were used for each experiment’s quantitative analysis. Final data were analyzed in GraphPad Prism using an unpaired *t*-test or ANOVA to determine statistical significance.

### 2.6. Cell Viability Assay

To evaluate the effects of MSI2 protein level on cell proliferation, we plated cells at a concentration of 500 cells/well in 96-well cell culture plates in triplicate. After 72 h of incubation to allow cells to grow, we added reagents for the CellTiter-Blue^®^ assay (Promega, Fitchburg, WI, USA) based on the manufacturer’s protocol to measure OD 562 nm.

### 2.7. Reverse Transcription and qPCR

RNA was extracted using a phenol–chloroform-based method. RNA concentration and quantity were measured using NanoDrop Lite (cat# ND-LITE ThermoFisher Scientific, Waltham, MA, USA). First-strand cDNA synthesis was performed using the iScript cDNA synthesis kit (cat#1708841, Biorad, CA, USA) according to the manufacturer’s instructions. The generated cDNA was diluted tenfold and used as a template for qPCR, which was performed with an Applied Biosystems QuantStudio 3 system using PowerTrack™ SYBR Green Master Mix (Applied Biosystems, Waltham, MA, USA). Relative quantification of gene expression was performed using the 2^−ΔΔCt^ method using the primers indicated in Appendix A.

### 2.8. ELISA

The concentration of VEGF-A in human lung cancer cell lines was determined by a VEGF Human ELISA kit (Abcam, #ab100662). ELISA assays were performed according to the manufacturer’s instructions. Briefly, the collected condition media from cells were added to a well coated with primary antibody and then immunosorbented using a biotinylated primary antibody at room temperature for 2.5 h. The color development catalyzed by horseradish peroxidase was terminated with 2.5 mol/L sulfuric acid and the absorption was measured at 450 nm. The protein concentration was determined by comparing the relative absorbance of the samples with the standards.

### 2.9. RNA-IP Assays

RNA-IP was performed using the Magna RIP RNA-binding Protein Immunoprecipitation kit (cat#17-700, Millipore, Burlington, MA, USA) according to the manufacturer’s protocol. Immunoprecipitated RNAs were quantified by quantitative PCR (qPCR) using the primers indicated in Appendix A, with PTP4A1 as a normalization (positive) control and GAPDH as a negative control.

### 2.10. RPPA

The RPPA analysis procedure of 344SQ-SCR, 344SQ-m1, and 344SQ-m2 mouse cells was previously described [25].

### 2.11. Immunohistochemistry of Human NSCLC

Human non-squamous NSCLC patient samples from the Rоstov Research Institute’s Human Tissue Repository Facility (HTRF) and from the Republican Clinical Oncology Dispensary named after Prof. M. Z. Sigal (RCOD at Kazan) were used. At the time of tissue acquisition, patients provided Institutional Review Board (IRB)–approved informed consent for the storage of tissues and the review of deidentified clinical data. Clinical information (Appendix A) from the repository database was abstracted in an anonymized fashion. Tissue samples were stained for VEGFR2 and MSI2 proteins via an immunohistochemical (IHC) approach, and hematoxylin and eosin (H&E) stained sections were used for morphological evaluation purposes, with unstained sections used for IHC staining using standard methods. The sections were incubated overnight with primary antibodies to MSI2 (EP1305Y, Rabbit, 1:100, Abcam #ab76148), VEGF-A (VG-1, Mouse, 1:50, Abcam #ab1316), and VEGFR2 (55B11, Rabbit, 1:50, Cell signaling, Cat #2479) at 4 °C in a humidified slide chamber. As a negative control, the primary antibody was replaced with normal mouse/rabbit IgG to confirm the absence of specific staining. Immunodetection was performed using the Dako Envision+ polymer system and immunostaining was visualized with the chromogen 3,3′-diaminobenzidine. All slides were viewed with a Nikon Eclipse 50i microscope and photomicrographs were taken with an attached Nikon DS-Fi1 camera (Melville, NY, USA).

### 2.12. In Silico Evaluation of MSI2 Binding to VEGFR2, VEGF-A, and PTEN mRNAs

Human and murine genome sequences for EGFR were obtained from the UCSC Human Gene Sorter December 2013 (GRCh38/hg38) assembly and scanned for the Musashi binding motifs previously defined by Bennett et al. [26] (15 motifs with the highest *p* values) and Wang et al. [27] (8 motifs with the highest *p* values; Appendix A).

### 2.13. Statistical Analysis

All statistical analyses, including unpaired two-tailed *t*-tests, ANOVA analysis, and Spearman correlations, were performed in GraphPad Prism 9 (San Diego, CA, USA).

## 3. Results

### 3.1. Musashi-2 Regulates VEGFR2 mRNA and Protein Levels and Directly Binds VEGFR2 mRNA in Human NSCLC Cell Lines

Our published paper suggested that MSI2 depletion in murine lung cancer cell line 344SQ may decrease VEGFR2 protein levels based on RPPA data (Appendix A) [25]. To verify this result, we used our previously established KRAS/P53-driven 344SQ cell line with doxycycline (DOX)-inducible knockdown (KD) of MSI2 (sh1, sh2) and with constant MSI2 overexpression (OE) (MSI2) (Appendix A) and evaluated VEGFR2 and VEGF-A protein and mRNA levels (Appendix A). We found that MSI2 KD leads to a significant decrease in VEGFR2 protein levels in mouse cells, though not a decrease in mRNA levels. In addition, MSI2 OE leads to a mild increase in VEGF-A mRNA and protein levels, but not VEGFR2 protein and mRNA levels. Thus, we conclude that MSI2 may regulate VEGFR2 and VEGF-A in mouse cells. Next, based on previously published papers [19,36,37], we selected several NSCLC cell lines with significant VEGFR2 expression and examined VEGFR2 and VEGF-A protein levels using Western blot analysis (Figure 1A) and *VEGFR2* mRNA levels (Figure 1B). We selected cell lines with higher expression of VEGFR2 (H441, Hcc1171, and Calu-1) and the A549 cell line as a negative control/low VEGFR2 expression sample for further analysis. Previously, Bennett et al. [26], Wang et al. [27], and Nguyen et al. [28] in their papers showed that MSI2 recognizes and directly binds consensus sequences with the core motif UAG in the 3′-untranslated region (3′-UTR) of mRNAs, and as a result regulates the stability and/or translation of target mRNAs. Based on this published data, we performed in silico analysis (using Bennett and Wang data) and found that *VEGFR2* and *VEGF-A* mRNAs have predicted binding sites for MSI2 binding (Figure 1C, Appendix A). To confirm the in silico results, we performed RNA immunoprecipitation (RIP) assays with an MSI2 antibody pulldown coupled with RT-qPCR in three lung adenocarcinoma cell lines: A549 (Appendix A), Hcc1171, and H441 (Figure 1D), using the previously defined MSI2 target mRNAs (*PTP4A*, *TGFβR1*, *SMAD3*) [25,38] as positive controls and *ACTB* and *GAPDH* as negative controls. Antibodies to MSI2 specifically immunoprecipitated the *VEGFR2* mRNA as efficiently as they did for the positive controls (Figure 1D). Taken together, we conclude that MSI2 directly regulates VEGFR2 mRNA translation in human lung adenocarcinoma.

### 3.2. Musashi-2 Regulation of VEGFR2 and VEGF-A Protein Levels in NSCLC

Since MSI2 directly binds *VEGFR2* mRNA in human lung adenocarcinoma cell lines, we established cell lines with MSI2 depletion and overexpression to evaluate the effect of MSI2 expression on VEGFR2 and VEGF-A protein levels. Depletion of MSI2 in human NSCLC cell lines led to a significant decrease in VEGFR2 protein levels and had a mixed effect on intracellular VEGF-A protein levels (Figure 2A,B, Appendix A). Additionally, MSI2 OE leads to slight increase in VEGFR2 protein levels in the A549 and H441 cell lines and a significant decrease in VEFR2 protein levels in the Hcc1171 and Calu-1 cell lines (Appendix A). In addition, we performed ELISA analysis of extracellular VEGF-A concentration after MSI2 depletion and overexpression (Figure 2C). ELISA analysis indicated that MSI2 depletion only leads to a decrease in VEGF-A in the A549 cell line, while MSI2 OE leads to a significant increase in VEGF-A concentration in cell lines that have moderate to high expression of VEGFR2 (H441, Hcc1171, and Calu-1). Moreover, RT-qPCR analysis of *VEGFR2* and *VEGF-A* with MSI2 depletion and overexpression showed that *VEGFR2* mRNA is significantly decreased in human lung adenocarcinoma cell lines with MSI2 depletion and increased with MSI2 OE in cell lines with moderate to high VEGFR2 expression (H441, Hcc1171, and Calu-1) (Appendix A). In parallel, evaluation of the effect of MSI2 depletion on cell growth in these cell lines showed that MSI2 level does not affect NSCLC cell growth (Appendix A). Therefore, we conclude that MSI2 positively regulates both *VEGFR2* mRNA and protein levels and also regulates *VEGF-A* mRNA translation in human lung cancer cells.

### 3.3. Musashi-2 Regulates AKT Signaling via PTEN mRNA Binding Independent of VEGFR2

Since MSI2 depletion leads to a reduction in VEGFR2 protein levels, we evaluated VEGFR2 downstream signaling in human NSCLC cell lines. Western blot analysis showed that MSI2 depletion reduces VEGFR2 and phAKT protein levels (Figure 3A,B, Appendix A). To assess the role of VEGFR2 in the downregulation of phAKT with MSI2 KD, we established A549 and H441 cell lines with inducible MSI2 KD and constant VEGFR2 overexpression (OE). Analysis of AKT signaling indicated that MSI2 KD leads to a decrease in phAKT and its downstream target phGSK3⍺/β with and without VEGFR2 OE of VEGFR2 (Figure 3C,D). These findings suggest that a decrease in VEGFR2 protein level does not affect AKT signaling.

Wang et al. [27] showed that MSI2 inhibits PTEN (tumor suppressor and inhibitor of AKT signaling) protein level in murine intestinal epithelia. Therefore, we evaluated PTEN protein level in human lung adenocarcinoma cell lines upon MSI2 depletion (Figure 3E,F). Western blot analysis showed significant upregulation of PTEN with MSI2 KD regardless of VEGFR2 level. This suggests that MSI2 may directly regulate *PTEN* mRNA levels and affect AKT signaling in lung adenocarcinoma. To test this, we performed in silico analysis and found that the 3′ UTR of *PTEN* mRNA has predicted binding sites for MSI2 (Figure 1D, Appendix A). Next, we performed RIP-qPCR analysis using an MSI2 antibody, which supported the in silico predictions and showed that MSI2 directly binds *PTEN* mRNA. Taken together, we conclude that MSI2 affects AKT signaling via direct *PTEN* mRNA binding in lung adenocarcinoma, independent of *VEGFR2* mRNA regulation.

### 3.4. Correlation of MSI2 with VEGFR2 and VEGF-A Expression in Human NSCLC

To evaluate the relationship between MSI2, VEGFR2, and VEGF-A in the subset of lung adenocarcinoma, we performed IHC analysis of MSI2, VEGFR2, and VEGF-A expression in an independent group of 116 non-squamous clinical tumor samples (Figure 4A,B, Appendix A). Spearman’s analysis of H scores showed a strong positive correlation for MSI2 vs. VEGFR2 (r = 0.673) and a moderate positive correlation for MSI2 vs. VEGF-A (r = 0.533).

## 4. Discussion

Published papers show that VEGFR2 and VEGF-A proteins are widely expressed by various tumor types, including lung tumors [16,17,18,19]. Additionally, *VEGFR2* overexpression is associated with chemoresistance and poor survival in patients with lung cancer [20]. Our study shows for the first time that MSI2 positively regulates VEGFR2 protein levels in NSCLC.

First, we selected human lung adenocarcinoma cell lines with high levels of *VEGFR2* expression (H441, Hcc1171, and Calu-1) and A549 as a control with a low level of VEGFR2 expression (Figure 1A,B). MSI2 typically regulates its targets by directly binding to specific motifs in the 3′-UTR fragments of mRNAs [26,27,28]. Therefore, we performed in silico analysis and then validated it through RNA-IP analysis (Figure 1C,D). In silico analysis and RNA-IP validation showed that MSI2 directly binds *VEGFR2* but not *VEGF-A* mRNA. These results expand our knowledge about the role of Musashi-2 in NSCLC progression.

We then established human NSCLC cell lines with MSI2 KD and MSI2 OE to see how MSI2 levels affect VEGFR2 and VEGF-A protein and mRNA levels (Figure 2, Appendix A). We found that MSI2 depletion leads not only to a decrease in VEGFR2 protein levels, but also a decrease in mRNA levels. That result can be interpreted in such a way that MSI2 regulates not only protein translation, but also affects mRNA transcript stability [29]. Moreover, we found that MSI2 overexpression leads to a modest increase in extracellular VEGF-A from cells with *VEGFR2* overexpression. This effect may be interpreted as the indirect positive regulation of VEGF-A by MSI2 [39,40].

As MSI2 regulates VEGFR2 protein levels, we evaluated cell proliferation along with VEGFR2 downstream signaling in human lung adenocarcinoma cell lines upon MSI2 depletion. The viability assay showed that MSI2 does not affect cell growth in these KRAS-driven cells (Appendix A). Signaling evaluations indicated a decrease in VEGFR2 and phAKT protein levels with MSI2 KD (Figure 3A,B), while phERK protein levels were not changed (Appendix A). This result is expected because our models involve *KRASmut*, and published papers show that *KRASmut* results in constitutive activation of ERK [41,42]. Next, we established cell lines with VEGFR2 OE to evaluate the effects of MSI2 on AKT signaling via VEGFR2 regulation. Our results indicated that MSI2 KD induces a decrease in phAKT and its downstream target GSK3⍺/β regardless of the VEGFR2 level (Figure 3C,D).

Wang et al. [27] previously showed that MSI2 inhibits PTEN protein levels in murine intestinal epithelia. Based on that, the evaluation of PTEN levels with MSI2 KD and VEGFR2 OE indicated that PTEN protein levels are increased upon MSI2 KD regardless of the VEGFR2 level (Figure 3E,F). In addition, we showed that MSI2 negatively directly regulates PTEN protein levels (Figure 1C,D, Appendix A). We next performed in silico analysis, which predicted MSI2 binding sites in PTEN mRNA, and such binding was validated using RNA IP/QPCR analysis. Taken together, our data suggest that PTEN is a direct MSI2 target in NSCLC. Therefore, MSI2 affects AKT signaling via direct *PTEN* mRNA binding, independent of *VEGFR2* mRNA regulation.

In addition, TMA analysis of human lung adenocarcinoma samples showed a positive correlation between MSI2 and VEGFR2 and VEGF-A protein levels (Figure 4). Taken together, we conclude that the previously unsuspected novel Musashi-2/VEGFR2 signaling axis is worth additional investigations and could be targeted for better NSCLC control in the future.

## 5. Conclusions

MSI2 protein directly regulates VEGFR2 and PTEN protein levels via *VEGFR2* and *PTEN* mRNA binding in lung adenocarcinoma. MSI2 protein expression was correlated with VEGFR2 and VEGF-A protein levels in non-squamous NSCLC patient samples. MSI2 is a promising therapeutic target for better control of NSCLC.

## Figures and Tables

**Figure 1 cancers-15-02529-f001:**
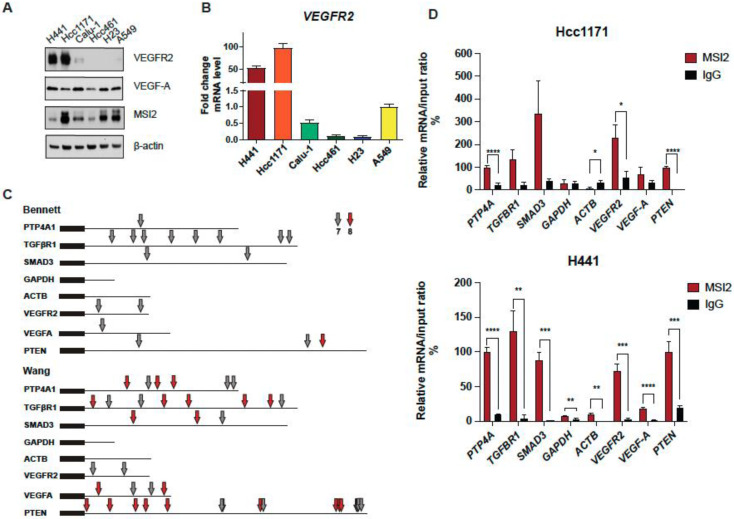
MSI2 directly binds VEGFR2 mRNA in human NSCLC. (**A**) Western blot representing images of human NSCLC cell lines. (**B**) VEGFR2 mRNA level in human NSCLC cell lines measured through RT-qPCR analysis. Data from at least three independent experiments normalized to 18S rRNA and to A549. (**C**) Predicted MSI2 binding sites in human mRNAs. Location of consensus binding sites for Musashi proteins in the noted human genes, as defined from studies by Bennett et al. and Wang et al. Coding sequences are thick lines and 3′-untranslated regions are thin lines. The 7- or 8-bp consensus sequences are indicated by arrows. *VEGFR2* reference sequence–NCBI Reference Sequence: NM_002253.4; *PTEN* reference sequence–NCBI Reference Sequence: NM_000314.8. (**D**) mRNA immunoprecipitation (RIP) analysis of indicated cell lines lysates using antibodies to MSI2 or IgG (negative control, followed by qRT-PCR). Data are normalized to positive control *PTP4A1*, *TGFBR1* and *SMAD3* are additional positive controls, and *GAPDH* and *ACTB* are negative controls. The data shown reflect the average of three independent RIP experiments. Error bars represented by SEM. Statistical analysis was performed using an unpaired two-tailed *t*-test. * *p* < 0.05, ** *p* < 0.01, *** *p* < 0.001, **** *p* < 0.0001 for all graphs.

**Figure 2 cancers-15-02529-f002:**
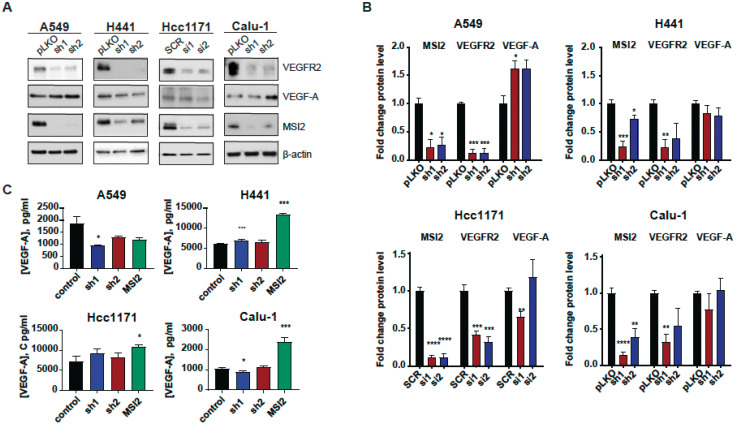
MSI2 regulation of VEGFR2 and VEGF-A protein levels in human NSCLC cell lines. (**A**) Representative Western blot images of NSCLC cell lines after MSI2 depletion by shRNA (sh1, sh2) and siRNA (si1, si2). Negative controls include pLKO and SCR. (**B**) Western blot quantifications of data from (**A**) were performed from at least three independent experiments using Image J software (version 1.53e), with values normalized to negative controls and β-actin. (**C**) The concentration of VEGF-A in cell culture media from the indicated cell lines following depletion by shRNA (sh1, sh2) and siRNA (si1, si2) and overexpression (MSI2) of MSI2. The ELISA data shown reflect the average of three independent experiments. shRNA KD of MSI2 was induced by the addition of 1 μg/mL of doxycycline for 48 h. Error bars represented by SEM. Statistical analysis was performed using an unpaired two-tailed *t*-test. * *p* < 0.05, ** *p* < 0.01, *** *p* < 0.001, **** *p* < 0.0001 for all graphs.

**Figure 3 cancers-15-02529-f003:**
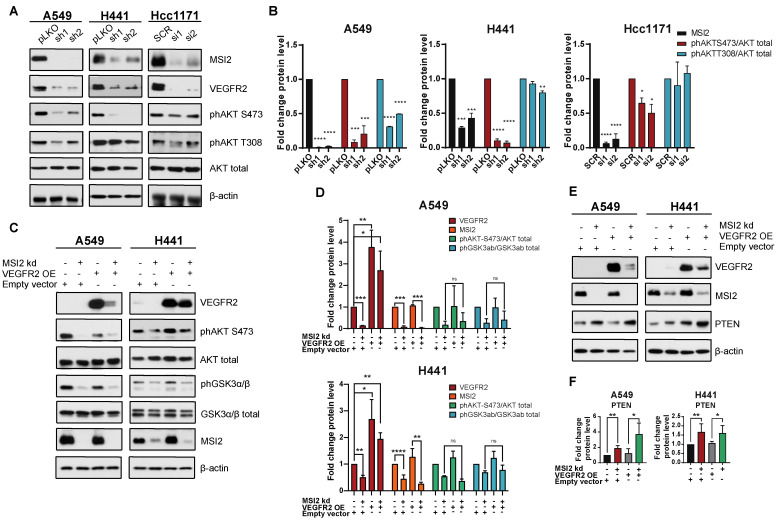
VEGFR2-independent effects of MSI2 KD on AKT signaling. (**A**) Representative Western blot images of NSCLC cell lines after MSI2 depletion by shRNA (sh1, sh2) and siRNA (si1, si2). Negative controls include pLKO and SCR. Negative controls include pLKO and SCR. (**B**) Quantification of Western blot data from (**A**) with values normalized to negative controls (pLKO, SCR) and β-actin. (**C**) Rescue experiment: representative Western blot images of A549 and H441 cell lines with MSI2 depletion by shRNA (sh1) and VEGFR2 overexpression (VEGFR2 OE). Negative control is an empty vector (pHAGE). (**D**) Quantification of Western blot data from (**C**) with values normalized to the empty vector and β-actin. (**E**) Western blot of indicated cell lines following depletion by shRNA (sh1) of MSI2 and VEGFR2 overexpression (VEGFR2 OE). (**F**) Quantification of Western blot data from (**E**) with values normalized to the empty vector and β-actin. Data from (**B**–**D**) were quantified from at least three independent experiments using Image J software. Error bars represented by SEM. Statistical analysis was performed using an unpaired two-tailed *t*-test. * *p* < 0.05, ** *p* < 0.01, *** *p* < 0.001, **** *p* < 0.0001.

**Figure 4 cancers-15-02529-f004:**
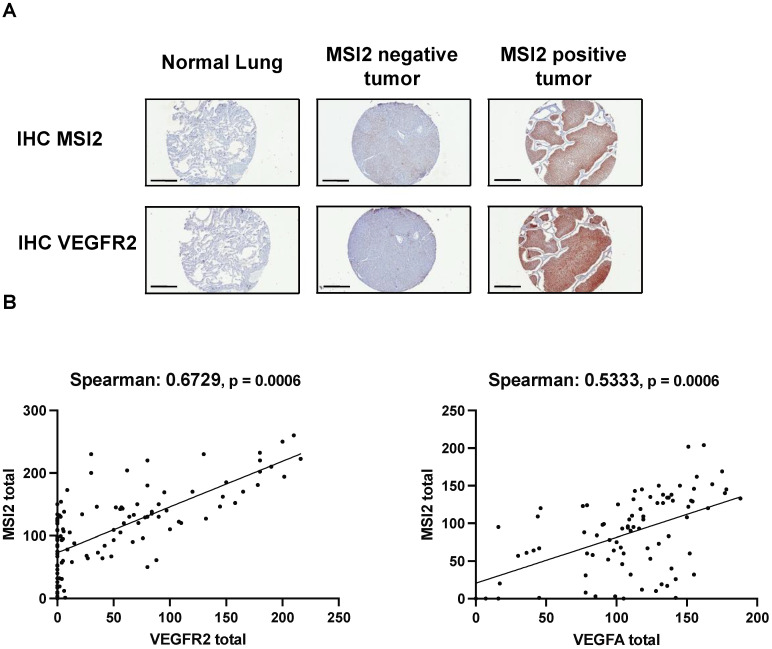
Expression of MSI2, VEGFR2, and VEGF-A proteins in human non-squamous NSCLC patient samples. (**A**) Representative IHC images of MSI2 and VEGFR2 expression in human normal lungs and lung tumors. (**B**) MSI2 and VEGF-A (n = 94) and MSI2 and VEGFR2 (n = 116) H score correlation in human NSCLC TMAs.

## Data Availability

The data presented in this study are available on request from the corresponding author and available to public.

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
