# Peer review of "Regulation of VEGFR2 and AKT Signaling by Musashi-2 in Lung Cancer"

_cancers, 2023, doi:10.3390/cancers15092529_

Round 1

Reviewer 1 Report

I have read and reviewed the manuscript “Regulation of VEGFR2 and AKT signaling by Musashi-2 in lung cancer”. This is a well-designed study that does explore regulation of VEGFR2 and AKT signaling by MSI-2. I have only one real concern that I bring up below. This point raises the question if the conclusions of this paper are robust enough for both squamous cell carcinoma and adenocarcinoma based upon the experiments and validation with the patient samples.

Major Points:

1.     In figure 3C, the gels for VEGFR2 both short and long exposure should be adjusted to not just show a large indistinguishable blot.

2.     In the supplemental data of the ~140 patients there was a combination of adenocarcinoma and squamous cell carcinoma histology. Did results of mRNA and protein expression differ by these two subtypes? Was this concept explored? I ask as I believe A549, Calu-1, and H23 are all adenocarcinoma cell lines.

3.     If all the in-vitro experiments were on adenocarcinoma, can we extrapolate the results to squamous cell carcinomas? If none of the cell lines were squamous cell carcinoma, then consideration for excluding all patients samples that were squamous cell carcinoma. This will help to identify the role of VEGFR2 and AKT signaling in adenocarcinoma.

4.     What were the histology of Hcc1171 and Hcc461?

Minor Points:

1.     Line 49, the first word is likely supposed to be “In the canonical model…”

2.     Line 119, consider “Details of this procedure were previously described [24].”

3.     Lines 189-190, should the sentence be referenced to your original paper?

Author Response

Response: We have attempted to experimentally address each of the points made.

Reviewer #1

I have read and reviewed the manuscript “Regulation of VEGFR2 and AKT signaling by Musashi-2 in lung cancer”. This is a well-designed study that does explore regulation of VEGFR2 and AKT signaling by MSI-2. I have only one real concern that I bring up below. This point raises the question if the conclusions of this paper are robust enough for both squamous cell carcinoma and adenocarcinoma based upon the experiments and validation with the patient samples.

Major Points:

  1. In figure 3C, the gels for VEGFR2 both short and long exposure should be adjusted to not just show a large indistinguishable blot.

Response: We updated Figure 3 based on your recommendations.

  1. In the supplemental data of the ~140 patients there was a combination of adenocarcinoma and squamous cell carcinoma histology. Did results of mRNA and protein expression differ by these two subtypes? Was this concept explored? I ask as I believe A549, Calu-1, and H23 are all adenocarcinoma cell lines.

Response: In our project we used lung adenocarcinoma cell lines. We didn’t evaluate and compare mRNAs levels in both subtypes, because the main concept is to show role of MSI2 in VEGFR2 and AKT signaling in lung adenocarcinoma. Because in our experiments we used lung adenocarcinoma cell lines, and because we only had few squamous cell carcinoma clinical samples, we decided to exclude squamous cell carcinoma data from our paper; we have now 116 total sample number in the TMA analysis. Figure 4B was revised accordingly.

  1. If all the in-vitro experiments were on adenocarcinoma, can we extrapolate the results to squamous cell carcinomas? If none of the cell lines were squamous cell carcinoma, then consideration for excluding all patients’ samples that were squamous cell carcinoma. This will help to identify the role of VEGFR2 and AKT signaling in adenocarcinoma.

Response: Thank you for noticing this discrepancy. Our work is based on lung adenocarcinoma cell lines. Therefore, we decided to exclude squamous cell carcinoma data from our paper. While MSI2 may have the same biological function in squamous cell carcinoma, additional experiments are needed with squamous cell carcinoma cell lines and tumor samples, that are beyond the scope of this paper.  

  1. What were the histology of Hcc1171 and Hcc461?

Response: Both Hcc1171 and Hcc461 cell lines have adenocarcinoma histologic types, and we clarified histology on line 86.

Minor Points:

  1. Line 49, the first word is likely supposed to be “In the canonical model…”
  2. Line 119, consider “Details of this procedure were previously described [24].”
  3. Lines 189-190, should the sentence be referenced to your original paper?

Response: We made all the corrections based on your minor points.

Sincerely,

Yanis Boumber                  

Reviewer 2 Report

This manuscript explored the mechnism of MSI2 in the Non-small cell lung cancer. The topic is interesting and the obtained results are useful and solid. I suggest to accept this manuscript as its current form.

Author Response

(The authors gave the same response as above.)
